# Safe and efficient off-policy reinforcement learning

**Rémi Munos**
munos@google.com
Google DeepMind

**Thomas Stepleton**
stepleton@google.com
Google DeepMind

**Anna Harutyunyan**
anna.harutyunyan@vub.ac.be
Vrije Universiteit Brussel

**Marc G. Bellemare**
bellemare@google.com
Google DeepMind

## Abstract

In this work, we take a fresh look at some old and new algorithms for off-policy, return-based reinforcement learning. Expressing these in a common form, we derive a novel algorithm, Retrace($\lambda$), with three desired properties: (1) it has *low variance*; (2) it *safely* uses samples collected from any behaviour policy, whatever its degree of "off-policyness"; and (3) it is *efficient* as it makes the best use of samples collected from near on-policy behaviour policies. We analyze the contractive nature of the related operator under both off-policy policy evaluation and control settings and derive online sample-based algorithms. We believe this is the first return-based off-policy control algorithm converging a.s. to $Q^*$ without the GLIE assumption (Greedy in the Limit with Infinite Exploration). As a corollary, we prove the convergence of Watkins' Q($\lambda$), which was an open problem since 1989. We illustrate the benefits of Retrace($\lambda$) on a standard suite of Atari 2600 games.

One fundamental trade-off in reinforcement learning lies in the definition of the update target: should one estimate Monte Carlo returns or bootstrap from an existing Q-function? Return-based methods (where *return* refers to the sum of discounted rewards $\sum_t \gamma^t r_t$) offer some advantages over value bootstrap methods: they are better behaved when combined with function approximation, and quickly propagate the fruits of exploration (Sutton, 1996). On the other hand, value bootstrap methods are more readily applied to off-policy data, a common use case. In this paper we show that *learning from returns need not be at cross-purposes with off-policy learning.*

We start from the recent work of Harutyunyan et al. (2016), who show that naive off-policy policy evaluation, without correcting for the "off-policyness" of a trajectory, still converges to the desired $Q^\pi$ value function provided the behavior $\mu$ and target $\pi$ policies are not too far apart (the maximum allowed distance depends on the $\lambda$ parameter). Their $Q^\pi(\lambda)$ algorithm learns from trajectories generated by $\mu$ simply by summing discounted off-policy corrected rewards at each time step. Unfortunately, the assumption that $\mu$ and $\pi$ are close is restrictive, as well as difficult to uphold in the control case, where the target policy is greedy with respect to the current Q-function. In that sense this algorithm is not *safe*: it does not handle the case of arbitrary "off-policyness".

Alternatively, the Tree-backup (TB($\lambda$)) algorithm (Precup et al., 2000) tolerates arbitrary target/behavior discrepancies by scaling information (here called *traces*) from future temporal differences by the product of target policy probabilities. TB($\lambda$) is not *efficient* in the "near on-policy" case (similar $\mu$ and $\pi$), though, as traces may be cut prematurely, blocking learning from full returns.

In this work, we express several off-policy, return-based algorithms in a common form. From this we derive an improved algorithm, Retrace($\lambda$), which is both *safe* and *efficient*, enjoying convergence guarantees for off-policy policy evaluation and – more importantly – for the control setting.

Retrace($\lambda$) can learn from full returns retrieved from past policy data, as in the context of experience replay (Lin, 1993), which has returned to favour with advances in deep reinforcement learning (Mnih et al., 2015; Schaul et al., 2016). Off-policy learning is also desirable for exploration, since it allows the agent to deviate from the target policy currently under evaluation.

To the best of our knowledge, this is the first online return-based off-policy control algorithm which does not require the GLIE (Greedy in the Limit with Infinite Exploration) assumption (Singh et al., 2000). In addition, we provide as a corollary the first proof of convergence of Watkins' Q($\lambda$) (see, e.g., Watkins, 1989; Sutton and Barto, 1998).

Finally, we illustrate the significance of Retrace($\lambda$) in a deep learning setting by applying it to the suite of Atari 2600 games provided by the Arcade Learning Environment (Bellemare et al., 2013).

# 1  Notation

We consider an agent interacting with a Markov Decision Process $(\mathcal{X}, \mathcal{A}, \gamma, P, r)$. $\mathcal{X}$ is a finite state space, $\mathcal{A}$ the action space, $\gamma \in [0, 1)$ the discount factor, $P$ the transition function mapping state-action pairs $(x, a) \in \mathcal{X} \times \mathcal{A}$ to distributions over $\mathcal{X}$, and $r : \mathcal{X} \times \mathcal{A} \to [-R_{\text{MAX}}, R_{\text{MAX}}]$ is the reward function. For notational simplicity we will consider a finite action space, but the case of infinite – possibly continuous – action space can be handled by the Retrace($\lambda$) algorithm as well. A policy $\pi$ is a mapping from $\mathcal{X}$ to a distribution over $\mathcal{A}$. A Q-function $Q$ maps each state-action pair $(x, a)$ to a value in $\mathbb{R}$; in particular, the reward $r$ is a Q-function. For a policy $\pi$ we define the operator $P^\pi$:

$$(P^\pi Q)(x, a) := \sum_{x' \in \mathcal{X}} \sum_{a' \in \mathcal{A}} P(x' \,|\, x, a)\pi(a' \,|\, x')Q(x', a').$$

The value function for a policy $\pi$, $Q^\pi$, describes the expected discounted sum of rewards associated with following $\pi$ from a given state-action pair. Using operator notation, we write this as

$$Q^\pi := \sum_{t \geq 0} \gamma^t (P^\pi)^t r. \tag{1}$$

The *Bellman operator* $\mathcal{T}^\pi$ for a policy $\pi$ is defined as $\mathcal{T}^\pi Q := r + \gamma P^\pi Q$ and its fixed point is $Q^\pi$, i.e. $\mathcal{T}^\pi Q^\pi = Q^\pi = (I - \gamma P^\pi)^{-1} r$. The *Bellman optimality operator* introduces a maximization over the set of policies:

$$\mathcal{T}Q := r + \gamma \max_\pi P^\pi Q. \tag{2}$$

Its fixed point is $Q^*$, the unique *optimal value function* (Puterman, 1994). It is this quantity that we will seek to obtain when we talk about the "control setting".

**Return-based Operators:**  The $\lambda$-return extension (Sutton, 1988) of the Bellman operators considers exponentially weighted sums of $n$-steps returns:

$$\mathcal{T}^\pi_\lambda Q := (1 - \lambda) \sum_{n \geq 0} \lambda^n \left[ (\mathcal{T}^\pi)^{n+1} Q \right] = Q + (I - \lambda\gamma P^\pi)^{-1} (\mathcal{T}^\pi Q - Q),$$

where $\mathcal{T}^\pi Q - Q$ is the *Bellman residual* of $Q$ for policy $\pi$. Examination of the above shows that $Q^\pi$ is also the fixed point of $\mathcal{T}^\pi_\lambda$. At one extreme ($\lambda = 0$) we have the Bellman operator $\mathcal{T}^\pi_{\lambda=0} Q = \mathcal{T}^\pi Q$, while at the other ($\lambda = 1$) we have the policy evaluation operator $\mathcal{T}^\pi_{\lambda=1} Q = Q^\pi$ which can be estimated using Monte Carlo methods (Sutton and Barto, 1998). Intermediate values of $\lambda$ trade off estimation bias with sample variance (Kearns and Singh, 2000).

We seek to evaluate a *target policy* $\pi$ using trajectories drawn from a *behaviour policy* $\mu$. If $\pi = \mu$, we are *on-policy*; otherwise, we are *off-policy*. We will consider trajectories of the form:

$$x_0 = x, a_0 = a, r_0, x_1, a_1, r_1, x_2, a_2, r_2, \ldots$$

with $a_t \sim \mu(\cdot|x_t)$, $r_t = r(x_t, a_t)$ and $x_{t+1} \sim P(\cdot|x_t, a_t)$. We denote by $\mathcal{F}_t$ this sequence up to time $t$, and write $\mathbb{E}_\mu$ the expectation with respect to both $\mu$ and the MDP transition probabilities. Throughout, we write $\| \cdot \|$ for supremum norm.

## 2 Off-Policy Algorithms

We are interested in two related off-policy learning problems. In the *policy evaluation* setting, we are given a fixed policy $\pi$ whose value $Q^\pi$ we wish to estimate from sample trajectories drawn from a behaviour policy $\mu$. In the *control* setting, we consider a sequence of policies that depend on our own sequence of Q-functions (such as $\varepsilon$-greedy policies), and seek to approximate $Q^*$.

The general operator that we consider for comparing several return-based off-policy algorithms is:

$$\mathcal{R}Q(x,a) := Q(x,a) + \mathbb{E}_\mu\Big[\sum_{t\geq 0}\gamma^t\Big(\prod_{s=1}^{t}c_s\Big)\big(r_t + \gamma\mathbb{E}_\pi Q(x_{t+1},\cdot) - Q(x_t,a_t)\big)\Big], \qquad (3)$$

for some non-negative coefficients $(c_s)$, where we write $\mathbb{E}_\pi Q(x,\cdot) := \sum_a \pi(a|x)Q(x,a)$ and define $(\prod_{s=1}^{t}c_s) = 1$ when $t = 0$. By extension of the idea of eligibility traces (Sutton and Barto, 1998), we informally call the coefficients $(c_s)$ the *traces* of the operator.

**Importance sampling (IS):** $c_s = \frac{\pi(a_s|x_s)}{\mu(a_s|x_s)}$. Importance sampling is the simplest way to correct for the discrepancy between $\mu$ and $\pi$ when learning from off-policy returns (Precup et al., 2000, 2001; Geist and Scherrer, 2014). The off-policy correction uses the product of the likelihood ratios between $\pi$ and $\mu$. Notice that $\mathcal{R}Q$ defined in (3) with this choice of $(c_s)$ yields $Q^\pi$ for any $Q$. For $Q = 0$ we recover the basic IS estimate $\sum_{t\geq 0}\gamma^t\big(\prod_{s=1}^{t}c_s\big)r_t$, thus (3) can be seen as a variance reduction technique (with a baseline $Q$). It is well known that IS estimates can suffer from large – even possibly infinite – variance (mainly due to the variance of the product $\frac{\pi(a_1|x_1)}{\mu(a_1|x_1)}\cdots\frac{\pi(a_t|x_t)}{\mu(a_t|x_t)}$), which has motivated further variance reduction techniques such as in (Mahmood and Sutton, 2015; Mahmood et al., 2015; Hallak et al., 2015).

**Off-policy $Q^\pi(\lambda)$ and $Q^*(\lambda)$:** $c_s = \lambda$. A recent alternative proposed by Harutyunyan et al. (2016) introduces an off-policy correction based on a $Q$-baseline (instead of correcting the probability of the sample path like in IS). This approach, called $Q^\pi(\lambda)$ and $Q^*(\lambda)$ for policy evaluation and control, respectively, corresponds to the choice $c_s = \lambda$. It offers the advantage of avoiding the blow-up of the variance of the product of ratios encountered with IS. Interestingly, this operator contracts around $Q^\pi$ provided that $\mu$ and $\pi$ are sufficiently close to each other. Defining $\varepsilon := \max_x \|\pi(\cdot|x) - \mu(\cdot|x)\|_1$ the level of "off-policyness", the authors prove that the operator defined by (3) with $c_s = \lambda$ is a contraction mapping around $Q^\pi$ for $\lambda < \frac{1-\gamma}{\gamma\varepsilon}$, and around $Q^*$ for the worst case of $\lambda < \frac{1-\gamma}{2\gamma}$. Unfortunately, $Q^\pi(\lambda)$ requires knowledge of $\varepsilon$, and the condition for $Q^*(\lambda)$ is very conservative. Neither $Q^\pi(\lambda)$, nor $Q^*(\lambda)$ are safe as they do not guarantee convergence for arbitrary $\pi$ and $\mu$.

**Tree-backup, TB($\lambda$):** $c_s = \lambda\pi(a_s|x_s)$. The TB($\lambda$) algorithm of Precup et al. (2000) corrects for the target/behaviour discrepancy by multiplying each term of the sum by the product of target policy probabilities. The corresponding operator defines a contraction mapping for any policies $\pi$ and $\mu$, which makes it a safe algorithm. However, this algorithm is not efficient in the near on-policy case (where $\mu$ and $\pi$ are similar) as it unnecessarily cuts the traces, preventing it to make use of full returns: indeed we need not discount stochastic on-policy transitions (as shown by Harutyunyan et al.'s results about $Q^\pi$).

**Retrace($\lambda$):** $c_s = \lambda\min\left(1, \frac{\pi(a_s|x_s)}{\mu(a_s|x_s)}\right)$. Our contribution is an algorithm – Retrace($\lambda$) – that takes the best of the three previous algorithms. Retrace($\lambda$) uses an importance sampling ratio truncated at $1$. Compared to IS, it does not suffer from the variance explosion of the product of IS ratios. Now, similarly to $Q^\pi(\lambda)$ and unlike TB($\lambda$), it does not cut the traces in the on-policy case, making it possible to benefit from the full returns. In the off-policy case, the traces are safely cut, similarly to TB($\lambda$). In particular, $\min\left(1, \frac{\pi(a_s|x_s)}{\mu(a_s|x_s)}\right) \geq \pi(a_s|x_s)$: Retrace($\lambda$) does not cut the traces as much as TB($\lambda$). In the subsequent sections, we will show the following:

- For any traces $0 \leq c_s \leq \pi(a_s|x_s)/\mu(a_s|x_s)$ (thus including the Retrace($\lambda$) operator), the return-based operator (3) is a $\gamma$-contraction around $Q^\pi$, for *arbitrary* policies $\mu$ and $\pi$

- In the control case (where $\pi$ is replaced by a sequence of increasingly greedy policies) the online Retrace($\lambda$) algorithm converges a.s. to $Q^*$, without requiring the GLIE assumption.

- As a corollary, Watkins's Q($\lambda$) converges a.s. to $Q^*$.

| | Definition of $c_s$ | Estimation variance | Guaranteed convergence† | Use full returns (near on-policy) |
|---|---|---|---|---|
| Importance sampling | $\frac{\pi(a_s\|x_s)}{\mu(a_s\|x_s)}$ | High | for any $\pi, \mu$ | yes |
| $Q^\pi(\lambda)$ | $\lambda$ | Low | for $\pi$ close to $\mu$ | yes |
| TB($\lambda$) | $\lambda\pi(a_s\|x_s)$ | Low | for any $\pi, \mu$ | no |
| Retrace($\lambda$) | $\lambda \min\left(1, \frac{\pi(a_s\|x_s)}{\mu(a_s\|x_s)}\right)$ | Low | for any $\pi, \mu$ | yes |

Table 1: Properties of several algorithms defined in terms of the general operator given in (3). †Guaranteed convergence of the expected operator $\mathcal{R}$.

## 3 Analysis of Retrace($\lambda$)

We will in turn analyze both off-policy policy evaluation and control settings. We will show that $\mathcal{R}$ is a contraction mapping in both settings (under a mild additional assumption for the control case).

### 3.1 Policy Evaluation

Consider a fixed target policy $\pi$. For ease of exposition we consider a fixed behaviour policy $\mu$, noting that our result extends to the setting of sequences of behaviour policies $(\mu_k : k \in \mathbb{N})$.

Our first result states the $\gamma$-contraction of the operator (3) defined by any set of non-negative coefficients $c_s = c_s(a_s, \mathcal{F}_s)$ (in order to emphasize that $c_s$ can be a function of the whole history $\mathcal{F}_s$) under the assumption that $0 \leq c_s \leq \frac{\pi(a_s|x_s)}{\mu(a_s|x_s)}$.

**Theorem 1.** *The operator $\mathcal{R}$ defined by (3) has a unique fixed point $Q^\pi$. Furthermore, if for each $a_s \in \mathcal{A}$ and each history $\mathcal{F}_s$ we have $c_s = c_s(a_s, \mathcal{F}_s) \in \left[0, \frac{\pi(a_s|x_s)}{\mu(a_s|x_s)}\right]$, then for any Q-function Q*

$$\|\mathcal{R}Q - Q^\pi\| \leq \gamma\|Q - Q^\pi\|.$$

The following lemma will be useful in proving Theorem 1 (proof in the appendix).

**Lemma 1.** *The difference between $\mathcal{R}Q$ and its fixed point $Q^\pi$ is*

$$\mathcal{R}Q(x,a) - Q^\pi(x,a) = \mathbb{E}_\mu\left[\sum_{t\geq 1}\gamma^t\left(\prod_{i=1}^{t-1}c_i\right)\left(\left[\mathbb{E}_\pi[(Q-Q^\pi)(x_t,\cdot)] - c_t(Q-Q^\pi)(x_t,a_t)\right]\right)\right].$$

*Proof (Theorem 1).* The fact that $Q^\pi$ is the fixed point of the operator $\mathcal{R}$ is obvious from (3) since $\mathbb{E}_{x_{t+1}\sim P(\cdot|x_t,a_t)}\left[r_t + \gamma\mathbb{E}_\pi Q^\pi(x_{x+1},\cdot) - Q^\pi(x_t,a_t)\right] = (\mathcal{T}^\pi Q^\pi - Q^\pi)(x_t,a_t) = 0$, since $Q^\pi$ is the fixed point of $\mathcal{T}^\pi$. Now, from Lemma 1, and defining $\Delta Q := Q - Q^\pi$, we have

$$\mathcal{R}Q(x,a) - Q^\pi(x,a) = \sum_{t\geq 1}\gamma^t\mathop{\mathbb{E}}_{\substack{x_{1:t}\\a_{1:t}}}\left[\left(\prod_{i=1}^{t-1}c_i\right)\left(\left[\mathbb{E}_\pi\Delta Q(x_t,\cdot) - c_t\Delta Q(x_t,a_t)\right]\right)\right]$$

$$= \sum_{t\geq 1}\gamma^t\mathop{\mathbb{E}}_{\substack{x_{1:t}\\a_{1:t-1}}}\left[\left(\prod_{i=1}^{t-1}c_i\right)\left(\left[\mathbb{E}_\pi\Delta Q(x_t,\cdot) - \mathbb{E}_{a_t}[c_t(a_t,\mathcal{F}_t)\Delta Q(x_t,a_t)|\mathcal{F}_t]\right]\right)\right]$$

$$= \sum_{t\geq 1}\gamma^t\mathop{\mathbb{E}}_{\substack{x_{1:t}\\a_{1:t-1}}}\left[\left(\prod_{i=1}^{t-1}c_i\right)\sum_b\left(\pi(b|x_t) - \mu(b|x_t)c_t(b,\mathcal{F}_t)\right)\Delta Q(x_t,b)\right].$$

Now since $\pi(a|x_t) - \mu(a|x_t)c_t(b,\mathcal{F}_t) \geq 0$, we have that $\mathcal{R}Q(x,a) - Q^\pi(x,a) = \sum_{y,b} w_{y,b}\Delta Q(y,b)$, i.e. a linear combination of $\Delta Q(y,b)$ weighted by non-negative coefficients:

$$w_{y,b} := \sum_{t\geq 1}\gamma^t\mathop{\mathbb{E}}_{\substack{x_{1:t}\\a_{1:t-1}}}\left[\left(\prod_{i=1}^{t-1}c_i\right)\left(\pi(b|x_t) - \mu(b|x_t)c_t(b,\mathcal{F}_t)\right)\mathbb{I}\{x_t = y\}\right].$$

The sum of those coefficients is:

$$\sum_{y,b} w_{y,b} = \sum_{t\geq 1}\gamma^t \underset{\substack{x_{1:t}\\a_{1:t-1}}}{\mathbb{E}}\left[\left(\prod_{i=1}^{t-1}c_i\right)\sum_b\big(\pi(b|x_t) - \mu(b|x_t)c_t(b,\mathcal{F}_t)\big)\right]$$

$$= \sum_{t\geq 1}\gamma^t \underset{\substack{x_{1:t}\\a_{1:t-1}}}{\mathbb{E}}\left[\left(\prod_{i=1}^{t-1}c_i\right)\mathbb{E}_{a_t}[1 - c_t(a_t,\mathcal{F}_t)|\mathcal{F}_t]\right] = \sum_{t\geq 1}\gamma^t \underset{\substack{x_{1:t}\\a_{1:t}}}{\mathbb{E}}\left[\left(\prod_{i=1}^{t-1}c_i\right)(1 - c_t)\right]$$

$$= \mathbb{E}_\mu\left[\sum_{t\geq 1}\gamma^t\left(\prod_{i=1}^{t-1}c_i\right) - \sum_{t\geq 1}\gamma^t\left(\prod_{i=1}^{t}c_i\right)\right] = \gamma C - (C-1),$$

where $C := \mathbb{E}_\mu\left[\sum_{t\geq 0}\gamma^t\left(\prod_{i=1}^t c_i\right)\right]$. Since $C \geq 1$, we have that $\sum_{y,b} w_{y,b} \leq \gamma$. Thus $\mathcal{R}Q(x,a) - Q^\pi(x,a)$ is a sub-convex combination of $\Delta Q(y,b)$ weighted by non-negative coefficients $w_{y,b}$ which sum to (at most) $\gamma$, thus $\mathcal{R}$ is a $\gamma$-contraction mapping around $Q^\pi$. □

**Remark 1.** *Notice that the coefficient $C$ in the proof of Theorem 1 depends on $(x,a)$. If we write $\eta(x,a) := 1 - (1-\gamma)\mathbb{E}_\mu\left[\sum_{t\geq 0}\gamma^t(\prod_{s=1}^t c_s)\right]$, then we have shown that*

$$|\mathcal{R}Q(x,a) - Q^\pi(x,a)| \leq \eta(x,a)\|Q - Q^\pi\|.$$

*Thus $\eta(x,a) \in [0,\gamma]$ is a $(x,a)$-specific contraction coefficient, which is $\gamma$ when $c_1 = 0$ (the trace is cut immediately) and can be close to zero when learning from full returns ($\mathbb{E}_\mu[c_t] \approx 1$ for all t).*

## 3.2 Control

In the control setting, the single target policy $\pi$ is replaced by a sequence of policies $(\pi_k)$ which depend on $(Q_k)$. While most prior work has focused on strictly greedy policies, here we consider the larger class of *increasingly greedy* sequences. We now make this notion precise.

**Definition 1.** *We say that a sequence of policies $(\pi_k : k \in \mathbb{N})$ is* increasingly greedy *w.r.t. a sequence $(Q_k : k \in \mathbb{N})$ of Q-functions if the following property holds for all k: $P^{\pi_{k+1}}Q_{k+1} \geq P^{\pi_k}Q_{k+1}$.*

Intuitively, this means that each $\pi_{k+1}$ is at least as greedy as the previous policy $\pi_k$ for $Q_{k+1}$. Many natural sequences of policies are increasingly greedy, including $\varepsilon_k$-greedy policies (with non-increasing $\varepsilon_k$) and softmax policies (with non-increasing temperature). See proofs in the appendix.

We will assume that $c_s = c_s(a_s, \mathcal{F}_s) = c(a_s, x_s)$ is Markovian, in the sense that it depends on $x_s, a_s$ (as well as the policies $\pi$ and $\mu$) only but not on the full past history. This allows us to define the (sub)-probability transition operator

$$(P^{c\mu}Q)(x,a) := \sum_{x'}\sum_{a'}p(x'|x,a)\mu(a'|x')c(a',x')Q(x',a').$$

Finally, an additional requirement to the convergence in the control case, we assume that $Q_0$ satisfies $\mathcal{T}^{\pi_0}Q_0 \geq Q_0$ (this can be achieved by a pessimistic initialization $Q_0 = -R_{MAX}/(1-\gamma)$).

**Theorem 2.** *Consider an arbitrary sequence of behaviour policies $(\mu_k)$ (which may depend on $(Q_k)$) and a sequence of target policies $(\pi_k)$ that are increasingly greedy w.r.t. the sequence $(Q_k)$:*

$$Q_{k+1} = \mathcal{R}_k Q_k,$$

*where the return operator $\mathcal{R}_k$ is defined by (3) for $\pi_k$ and $\mu_k$ and a Markovian $c_s = c(a_s, x_s) \in [0, \frac{\pi_k(a_s|x_s)}{\mu_k(a_s|x_s)}]$. Assume the target policies $\pi_k$ are $\varepsilon_k$-away from the greedy policies w.r.t. $Q_k$, in the sense that $\mathcal{T}^{\pi_k}Q_k \geq \mathcal{T}Q_k - \varepsilon_k\|Q_k\|e$, where $e$ is the vector with 1-components. Further suppose that $\mathcal{T}^{\pi_0}Q_0 \geq Q_0$. Then for any $k \geq 0$,*

$$\|Q_{k+1} - Q^*\| \leq \gamma\|Q_k - Q^*\| + \varepsilon_k\|Q_k\|.$$

*In consequence, if $\varepsilon_k \to 0$ then $Q_k \to Q^*$.*

*Sketch of Proof (The full proof is in the appendix).* Using $P^{c\mu_k}$, the Retrace($\lambda$) operator rewrites

$$\mathcal{R}_k Q = Q + \sum_{t\geq 0}\gamma^t(P^{c\mu_k})^t(\mathcal{T}^{\pi_k}Q - Q) = Q + (I - \gamma P^{c\mu_k})^{-1}(\mathcal{T}^{\pi_k}Q - Q).$$

We now lower- and upper-bound the term $Q_{k+1} - Q^*$.

**Upper bound on $Q_{k+1} - Q^*$.** We prove that $Q_{k+1} - Q^* \le A_k(Q_k - Q^*)$ with $A_k := \gamma(I - \gamma P^{c\mu_k})^{-1}[P^{\pi_k} - P^{c\mu_k}]$. Since $c_t \in [0, \frac{\pi(a_t|x_t)}{\mu(a_t|x_t)}]$ we deduce that $A_k$ has non-negative elements, whose sum over each row, is at most $\gamma$. Thus

$$Q_{k+1} - Q^* \le \gamma \|Q_k - Q^*\| e. \tag{4}$$

**Lower bound on $Q_{k+1} - Q^*$.** Using the fact that $\mathcal{T}^{\pi_k}Q_k \ge \mathcal{T}^{\pi^*}Q_k - \varepsilon_k\|Q_k\|e$ we have

$$
\begin{aligned}
Q_{k+1} - Q^* &\ge\ Q_{k+1} - \mathcal{T}^{\pi_k}Q_k + \gamma P^{\pi^*}(Q_k - Q^*) - \gamma\varepsilon_k\|Q_k\|e \\
&=\ \gamma P^{c\mu_k}(I - \gamma P^{c\mu_k})^{-1}(\mathcal{T}^{\pi_k}Q_k - Q_k) + \gamma P^{\pi^*}(Q_k - Q^*) - \varepsilon_k\|Q_k\|e. 
\end{aligned} \tag{5}
$$

**Lower bound on $\mathcal{T}^{\pi_k}Q_k - Q_k$.** Since the sequence $(\pi_k)$ is increasingly greedy w.r.t. $(Q_k)$, we have

$$
\begin{aligned}
\mathcal{T}^{\pi_{k+1}}Q_{k+1} - Q_{k+1} &\ge\ \mathcal{T}^{\pi_k}Q_{k+1} - Q_{k+1} = r + (\gamma P^{\pi_k} - I)\mathcal{R}_k Q_k \\
&=\ B_k(\mathcal{T}^{\pi_k}Q_k - Q_k),
\end{aligned} \tag{6}
$$

where $B_k := \gamma[P^{\pi_k} - P^{c\mu_k}](I - \gamma P^{c\mu_k})^{-1}$. Since $P^{\pi_k} - P^{c\mu_k}$ and $(I - \gamma P^{c\mu_k})^{-1}$ are non-negative matrices, so is $B_k$. Thus $\mathcal{T}^{\pi_k}Q_k - Q_k \ge B_{k-1}B_{k-2}\ldots B_0(\mathcal{T}^{\pi_0}Q_0 - Q_0) \ge 0$, since we assumed $T^{\pi_0}Q_0 - Q_0 \ge 0$. Thus, (5) implies that

$$Q_{k+1} - Q^* \ge \gamma P^{\pi^*}(Q_k - Q^*) - \varepsilon_k\|Q_k\|e.$$

Combining the above with (4) we deduce $\|Q_{k+1} - Q^*\| \le \gamma\|Q_k - Q^*\| + \varepsilon_k\|Q_k\|$. When $\varepsilon_k \to 0$, we further deduce that $Q_k$ are bounded, thus $Q_k \to Q^*$. $\qquad\square$

### 3.3   Online algorithms

So far we have analyzed the contraction properties of the expected $\mathcal{R}$ operators. We now describe online algorithms which can learn from sample trajectories. We analyze the algorithms in the *every visit* form (Sutton and Barto, 1998), which is the more practical generalization of the first-visit form. In this section, we will only consider the Retrace($\lambda$) algorithm defined with the coefficient $c = \lambda\min(1, \pi/\mu)$. For that $c$, let us rewrite the operator $P^{c\mu}$ as $\lambda P^{\pi\wedge\mu}$, where $P^{\pi\wedge\mu}Q(x,a) := \sum_y\sum_b \min(\pi(b|y), \mu(b|y))Q(y,b)$, and write the Retrace operator $\mathcal{R}Q = Q + (I - \lambda\gamma P^{\pi\wedge\mu})^{-1}(\mathcal{T}^{\pi}Q - Q)$. We focus on the control case, noting that a similar (and simpler) result can be derived for policy evaluation.

**Theorem 3.** *Consider a sequence of sample trajectories, with the $k^{th}$ trajectory $x_0, a_0, r_0, x_1, a_1, r_1, \ldots$ generated by following $\mu_k$: $a_t \sim \mu_k(\cdot|x_t)$. For each $(x,a)$ along this trajectory, with $s$ being the time of first occurrence of $(x,a)$, update*

$$Q_{k+1}(x,a) \leftarrow Q_k(x,a) + \alpha_k\sum_{t\ge s}\delta_t^{\pi_k}\sum_{j=s}^{t}\gamma^{t-j}\Big(\prod_{i=j+1}^{t}c_i\Big)\mathbb{I}\{x_j, a_j = x, a\}, \tag{7}$$

*where $\delta_t^{\pi_k} := r_t + \gamma\mathbb{E}_{\pi_k}Q_k(x_{t+1},\cdot) - Q_k(x_t, a_t)$, $\alpha_k = \alpha_k(x_s, a_s)$. We consider the Retrace($\lambda$) algorithm where $c_i = \lambda\min\big(1, \frac{\pi(a_i|x_i)}{\mu(a_i|x_i)}\big)$. Assume that $(\pi_k)$ are increasingly greedy w.r.t. $(Q_k)$ and are each $\varepsilon_k$-away from the greedy policies $(\pi_{Q_k})$, i.e. $\max_x\|\pi_k(\cdot|x) - \pi_{Q_k}(\cdot|x)\|_1 \le \varepsilon_k$, with $\varepsilon_k \to 0$. Assume that $P^{\pi_k}$ and $P^{\pi_k\wedge\mu_k}$ asymptotically commute: $\lim_k\|P^{\pi_k}P^{\pi_k\wedge\mu_k} - P^{\pi_k\wedge\mu_k}P^{\pi_k}\| = 0$. Assume further that (1) all states and actions are visited infinitely often: $\sum_{t\ge 0}\mathbb{P}\{x_t, a_t = x, a\} \ge D > 0$, (2) the sample trajectories are finite in terms of the second moment of their lengths $T_k$: $\mathbb{E}_{\mu_k}T_k^2 < \infty$, (3) the stepsizes obey the usual Robbins-Munro conditions. Then $Q_k \to Q^*$ a.s.*

The proof extends similar convergence proofs of TD($\lambda$) by Bertsekas and Tsitsiklis (1996) and of optimistic policy iteration by Tsitsiklis (2003), and is provided in the appendix. Notice that compared to Theorem 2 we do not assume that $\mathcal{T}^{\pi_0}Q_0 - Q_0 \ge 0$ here. However, we make the additional (rather technical) assumption that $P^{\pi_k}$ and $P^{\pi_k\wedge\mu_k}$ commute at the limit. This is satisfied for example when the probability assigned by the behavior policy $\mu_k(\cdot|x)$ to the greedy action $\pi_{Q_k}(x)$ is independent of $x$. Examples include $\varepsilon$-greedy policies, or more generally mixtures between the greedy policy $\pi_{Q_k}$ and an arbitrary distribution $\mu$ (see Lemma 5 in the appendix for the proof):

$$\mu_k(a|x) = \varepsilon\frac{\mu(a|x)}{1 - \mu(\pi_{Q_k}(x)|x)}\mathbb{I}\{a \ne \pi_{Q_k}(x)\} + (1-\varepsilon)\mathbb{I}\{a = \pi_{Q_k}(x)\}. \tag{8}$$

Notice that the mixture coefficient $\varepsilon$ needs not go to 0.

# 4 Discussion of the results

## 4.1 Choice of the trace coefficients $c_s$

Theorems 1 and 2 ensure convergence to $Q^\pi$ and $Q^*$ for any trace coefficient $c_s \in [0, \frac{\pi(a_s|x_s)}{\mu(a_s|x_s)}]$. However, to make the best choice of $c_s$, we need to consider the *speed* of convergence, which depends on both (1) the variance of the online estimate, which indicates how many online updates are required in a single iteration of $\mathcal{R}$, and (2) the contraction coefficient of $\mathcal{R}$.

**Variance:** The variance of the estimate strongly depends on the variance of the product trace $(c_1 \ldots c_t)$, which is not an easy quantity to control in general, as the $(c_s)$ are usually not independent. However, assuming independence and stationarity of $(c_s)$, we have that $\mathbb{V}\left(\sum_t \gamma^t c_1 \ldots c_t\right)$ is at least $\sum_t \gamma^{2t} \mathbb{V}(c)^t$, which is finite only if $\mathbb{V}(c) < 1/\gamma^2$. Thus, an important requirement for a numerically stable algorithm is for $\mathbb{V}(c)$ to be as small as possible, and certainly no more than $1/\gamma^2$. This rules out importance sampling (for which $c = \frac{\pi(a|x)}{\mu(a|x)}$, and $\mathbb{V}(c|x) = \sum_a \mu(a|x)\left(\frac{\pi(a|x)}{\mu(a|x)} - 1\right)^2$, which may be larger than $1/\gamma^2$ for some $\pi$ and $\mu$), and is the reason we choose $c \leq 1$.

**Contraction speed:** The contraction coefficient $\eta \in [0, \gamma]$ of $\mathcal{R}$ (see Remark 1) depends on how much the traces have been cut, and should be as small as possible (since it takes $\log(1/\varepsilon)/\log(1/\eta)$ iterations of $\mathcal{R}$ to obtain an $\varepsilon$-approximation). It is smallest when the traces are not cut at all (i.e. if $c_s = 1$ for all $s$, $\mathcal{R}$ is the policy evaluation operator which produces $Q^\pi$ in a single iteration). Indeed, when the traces are cut, we do not benefit from learning from full returns (in the extreme, $c_1 = 0$ and $\mathcal{R}$ reduces to the (one step) Bellman operator with $\eta = \gamma$).

A reasonable trade-off between low variance (when $c_s$ are small) and high contraction speed (when $c_s$ are large) is given by Retrace($\lambda$), for which we provide the convergence of the online algorithm.

If we relax the assumption that the trace is Markovian (in which case only the result for policy evaluation has been proven so far) we could trade off a low trace at some time for a possibly larger-than-1 trace at another time, as long as their product is less than 1. A possible choice could be

$$c_s = \lambda \min\left(\frac{1}{c_1 \ldots c_{s-1}}, \frac{\pi(a_s|x_s)}{\mu(a_s|x_s)}\right). \tag{9}$$

## 4.2 Other topics of discussion

**No GLIE assumption.** The crucial point of Theorem 2 is that convergence to $Q^*$ occurs for *arbitrary* behaviour policies. Thus the online result in Theorem 3 does not require the behaviour policies to become greedy in the limit with infinite exploration (i.e. GLIE assumption, Singh et al., 2000). We believe Theorem 3 provides the first convergence result to $Q^*$ for a $\lambda$-return (with $\lambda > 0$) algorithm that does not require this (hard to satisfy) assumption.

**Proof of Watkins' Q($\lambda$).** As a corollary of Theorem 3 when selecting our target policies $\pi_k$ to be greedy w.r.t. $Q_k$ (i.e. $\varepsilon_k = 0$), we deduce that Watkins' Q($\lambda$) (e.g., Watkins, 1989; Sutton and Barto, 1998) converges a.s. to $Q^*$ (under the assumption that $\mu_k$ commutes asymptotically with the greedy policies, which is satisfied for e.g. $\mu_k$ defined by (8)). We believe this is the first such proof.

**Increasingly greedy policies** The assumption that the sequence of target policies $(\pi_k)$ is increasingly greedy w.r.t. the sequence of $(Q_k)$ is more general that just considering greedy policies w.r.t. $(Q_k)$ (which is Watkins's Q($\lambda$)), and leads to more efficient algorithms. Indeed, using non-greedy target policies $\pi_k$ may speed up convergence as the traces are not cut as frequently. Of course, in order to converge to $Q^*$, we eventually need the target policies (and not the behaviour policies, as mentioned above) to become greedy in the limit (i.e. $\varepsilon_k \to 0$ as defined in Theorem 2).

**Comparison to $Q^\pi(\lambda)$.** Unlike Retrace($\lambda$), $Q^\pi(\lambda)$ does not need to know the behaviour policy $\mu$. However, it fails to converge when $\mu$ is far from $\pi$. Retrace($\lambda$) uses its knowledge of $\mu$ (for the chosen actions) to cut the traces and safely handle arbitrary policies $\pi$ and $\mu$.

**Comparison to TB($\lambda$).** Similarly to $Q^\pi(\lambda)$, TB($\lambda$) does not need the knowledge of the behaviour policy $\mu$. But as a consequence, TB($\lambda$) is not able to benefit from possible near on-policy situations, cutting traces unnecessarily when $\pi$ and $\mu$ are close.

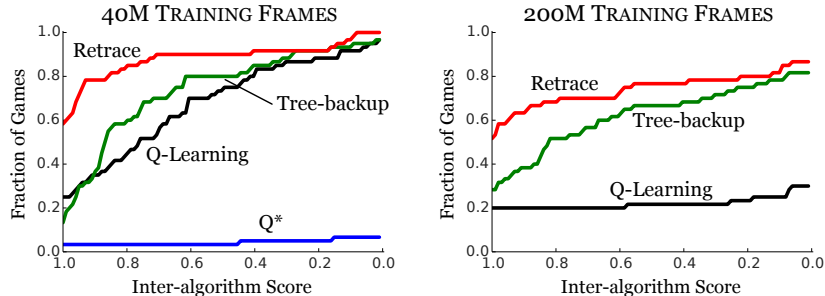

Figure 1: Inter-algorithm score distribution for $\lambda$-return ($\lambda = 1$) variants and Q-Learning ($\lambda = 0$).

**Estimating the behavior policy.** In the case $\mu$ is unknown, it is reasonable to build an estimate $\widehat{\mu}$ from observed samples and use $\widehat{\mu}$ instead of $\mu$ in the definition of the trace coefficients $c_s$. This may actually even lead to a better estimate, as analyzed by Li et al. (2015).

**Continuous action space.** Let us mention that Theorems 1 and 2 extend to the case of (measurable) continuous or infinite action spaces. The trace coefficients will make use of the densities $\min(1, d\pi/d\mu)$ instead of the probabilities $\min(1, \pi/\mu)$. This is not possible with TB($\lambda$).

**Open questions include:** (1) Removing the technical assumption that $P^{\pi_k}$ and $P^{\pi_k \wedge \mu_k}$ asymptotically commute, (2) Relaxing the Markov assumption in the control case in order to allow trace coefficients $c_s$ of the form (9).

## 5    Experimental Results

To validate our theoretical results, we employ Retrace($\lambda$) in an experience replay (Lin, 1993) setting, where sample transitions are stored within a large but bounded *replay memory* and subsequently replayed as if they were new experience. Naturally, older data in the memory is usually drawn from a policy which differs from the current policy, offering an excellent point of comparison for the algorithms presented in Section 2.

Our agent adapts the DQN architecture of Mnih et al. (2015) to replay short sequences from the memory (details in the appendix) instead of single transitions. The Q-function target update for a sample sequence $x_t, a_t, r_t, \cdots, x_{t+k}$ is

$$\Delta Q(x_t, a_t) = \sum_{s=t}^{t+k-1} \gamma^{s-t} \Big( \prod_{i=t+1}^{s} c_i \Big) \big[ r(x_s, a_s) + \gamma \mathbb{E}_\pi Q(x_{s+1}, \cdot) - Q(x_s, a_s) \big].$$

We compare our algorithms' performance on 60 different Atari 2600 games in the Arcade Learning Environment (Bellemare et al., 2013) using Bellemare et al.'s inter-algorithm score distribution. Inter-algorithm scores are normalized so that 0 and 1 respectively correspond to the worst and best score for a particular game, within the set of algorithms under comparison. If $g \in \{1, \ldots, 60\}$ is a game and $z_{g,a}$ the inter-algorithm score on $g$ for algorithm $a$, then the score distribution function is $f(x) := |\{g : z_{g,a} \geq x\}|/60$. Roughly, a strictly higher curve corresponds to a better algorithm.

Across values of $\lambda$, $\lambda = 1$ performs best, save for $Q^*(\lambda)$ where $\lambda = 0.5$ obtains slightly superior performance. However, is highly sensitive to the choice of $\lambda$ (see Figure 1, left, and Table 2 in the appendix). Both Retrace($\lambda$) and TB($\lambda$) achieve dramatically higher performance than Q-Learning early on and maintain their advantage throughout. Compared to TB($\lambda$), Retrace($\lambda$) offers a narrower but still marked advantage, being the best performer on 30 games; TB($\lambda$) claims 15 of the remainder. Per-game details are given in the appendix.

**Conclusion.** Retrace($\lambda$) can be seen as an algorithm that automatically adjusts – efficiently and safely – the length of the return to the degree of "off-policyness" of any available data.

**Acknowledgments.** The authors thank Daan Wierstra, Nicolas Heess, Hado van Hasselt, Ziyu Wang, David Silver, Audrunas Grūslys, Georg Ostrovski, Hubert Soyer, and others at Google Deep-Mind for their very useful feedback on this work.

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
