[Supplementary Material · retrace.pdf]

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

## A    Proof of Lemma 1

*Proof (Lemma 1).* Let $\Delta Q := Q - Q^\pi$. We begin by rewriting (3):

$$\mathcal{R}Q(x,a) = \sum_{t \ge 0} \gamma^t \mathbb{E}_\mu \left[ \left( \prod_{s=1}^{t} c_s \right) \left( r_t + \gamma \left[ \mathbb{E}_\pi Q(x_{t+1}, \cdot) - c_{t+1} Q(x_{t+1}, a_{t+1}) \right] \right) \right].$$

Since $Q^\pi$ is the fixed point of $\mathcal{R}$, we have

$$Q^\pi(x,a) = \mathcal{R}Q^\pi(x,a) = \sum_{t \ge 0} \gamma^t \mathbb{E}_\mu \left[ \left( \prod_{s=1}^{t} c_s \right) \left( r_t + \gamma \left[ \mathbb{E}_\pi Q^\pi(x_{t+1}, \cdot) - c_{t+1} Q^\pi(x_{t+1}, a_{t+1}) \right] \right) \right],$$

from which we deduce that

$$
\begin{aligned}
\mathcal{R}Q(x,a) - Q^\pi(x,a) &= \sum_{t \ge 0} \gamma^t \mathbb{E}_\mu \left[ \left( \prod_{s=1}^{t} c_s \right) \left( \gamma \left[ \mathbb{E}_\pi \Delta Q(x_{t+1}, \cdot) - c_{t+1} \Delta Q(x_{t+1}, a_{t+1}) \right] \right) \right] \\
&= \sum_{t \ge 1} \gamma^t \mathbb{E}_\mu \left[ \left( \prod_{s=1}^{t-1} c_s \right) \left( \left[ \mathbb{E}_\pi \Delta Q(x_t, \cdot) - c_t \Delta Q(x_t, a_t) \right] \right) \right].
\end{aligned}
$$

$\square$

## B    Increasingly greedy policies

Recall the definition of an increasingly greedy sequence of policies.

**Definition 2.** *We say that a sequence of policies $(\pi_k)$ is increasingly greedy w.r.t. a sequence of functions $(Q_k)$ if the following property holds for all $k$:*

$$P^{\pi_{k+1}} Q_{k+1} \ge P^{\pi_k} Q_{k+1}.$$

It is obvious to see that this property holds if all policies $\pi_k$ are greedy w.r.t. $Q_k$. Indeed in such case, $\mathcal{T}^{\pi_{k+1}} Q_{k+1} = \mathcal{T}Q_{k+1} \ge \mathcal{T}^\pi Q_{k+1}$ for any $\pi$.

We now prove that this property holds for $\varepsilon_k$-greedy policies (with non-increasing $(\varepsilon_k)$) as well as soft-max policies (with non-decreasing $(\beta_k)$), as stated in the two lemmas below.

Of course not all policies satisfy this property (a counter-example being $\pi_k(a|x) := \arg\min_{a'} Q_k(x, a')$).

**Lemma 2.** *Let $(\varepsilon_k)$ be a non-increasing sequence. Then the sequence of policies $(\pi_k)$ which are $\varepsilon_k$-greedy w.r.t. the sequence of functions $(Q_k)$ is increasingly greedy w.r.t. that sequence.*

*Proof.* From the definition of an $\varepsilon$-greedy policy we have:

$$
\begin{aligned}
P^{\pi_{k+1}} Q_{k+1}(x,a) &= \sum_y p(y|x,a) \left[ (1 - \varepsilon_{k+1}) \max_b Q_{k+1}(y,b) + \varepsilon_{k+1} \frac{1}{A} \sum_b Q_{k+1}(y,b) \right] \\
&\ge \sum_y p(y|x,a) \left[ (1 - \varepsilon_k) \max_b Q_{k+1}(y,b) + \varepsilon_k \frac{1}{A} \sum_b Q_{k+1}(y,b) \right] \\
&\ge \sum_y p(y|x,a) \left[ (1 - \varepsilon_k) Q_{k+1}(y, \arg\max_b Q_k(y,b)) + \varepsilon_k \frac{1}{A} \sum_b Q_{k+1}(y,b) \right] \\
&= P^{\pi_k} Q_{k+1},
\end{aligned}
$$

where we used the fact that $\varepsilon_{k+1} \le \varepsilon_k$. $\square$

**Lemma 3.** *Let $(\beta_k)$ be a non-decreasing sequence of soft-max parameters. Then the sequence of policies $(\pi_k)$ which are soft-max (with parameter $\beta_k$) w.r.t. the sequence of functions $(Q_k)$ is increasingly greedy w.r.t. that sequence.*

*Proof.* For any $Q$ and $y$, define $\pi_\beta(b) = \frac{e^{\beta Q(y,b)}}{\sum_{b'} e^{\beta Q(y,b')}}$ and $f(\beta) = \sum_b \pi_\beta(b) Q(y,b)$. Then we have

$$
\begin{aligned}
f'(\beta) &= \sum_b \Big[ \pi_\beta(b) Q(y,b) - \pi_\beta(b) \sum_{b'} \pi_\beta(b') Q(y,b') \Big] Q(y,b) \\
&= \sum_b \pi_\beta(b) Q(y,b)^2 - \Big( \sum_b \pi_\beta(b) Q(y,b) \Big)^2 \\
&= \mathbb{V}_{b \sim \pi_\beta} \big[ Q(y,b) \big] \geq 0.
\end{aligned}
$$

Thus $\beta \mapsto f(\beta)$ is a non-decreasing function, and since $\beta_{k+1} \geq \beta_k$, we have

$$
\begin{aligned}
P^{\pi_{k+1}} Q_{k+1}(x,a) &= \sum_y p(y|x,a) \sum_b \frac{e^{\beta_{k+1} Q_{k+1}(y,b)}}{\sum_{b'} e^{\beta_{k+1} Q_{k+1}(y,b')}} Q_{k+1}(y,b) \\
&\geq \sum_y p(y|x,a) \sum_b \frac{e^{\beta_k Q_{k+1}(y,b)}}{\sum_{b'} e^{\beta_k Q_{k+1}(y,b')}} Q_{k+1}(y,b) \\
&= P^{\pi_k} Q_{k+1}(x,a). \qquad \qquad \square
\end{aligned}
$$

## C  Proof of Theorem 2

As mentioned in the main text, since $c_s$ is Markovian, we can define the (sub)-probability transition operator

$$
(P^{c\mu} Q)(x,a) := \sum_{x'} \sum_{a'} p(x'|x,a) \mu(a'|x') c(a',x') Q(x',a').
$$

The Retrace($\lambda$) operator then writes

$$
\mathcal{R}_k Q = Q + \sum_{t \geq 0} \gamma^t (P^{c\mu_k})^t (\mathcal{T}^{\pi_k} Q - Q) = Q + (I - \gamma P^{c\mu_k})^{-1} (\mathcal{T}^{\pi_k} Q - Q).
$$

*Proof.* We now lower- and upper-bound the term $Q_{k+1} - Q^*$.

**Upper bound on $Q_{k+1} - Q^*$.** Since $Q_{k+1} = \mathcal{R}_k Q_k$, we have

$$
\begin{aligned}
Q_{k+1} - Q^* &= Q_k - Q^* + (I - \gamma P^{c\mu_k})^{-1} \big[ \mathcal{T}^{\pi_k} Q_k - Q_k \big] \\
&= (I - \gamma P^{c\mu_k})^{-1} \big[ \mathcal{T}^{\pi_k} Q_k - Q_k + (I - \gamma P^{c\mu_k})(Q_k - Q^*) \big] \\
&= (I - \gamma P^{c\mu_k})^{-1} \big[ \mathcal{T}^{\pi_k} Q_k - Q^* - \gamma P^{c\mu_k}(Q_k - Q^*) \big] \\
&= (I - \gamma P^{c\mu_k})^{-1} \big[ \mathcal{T}^{\pi_k} Q_k - \mathcal{T} Q^* - \gamma P^{c\mu_k}(Q_k - Q^*) \big] \\
&\leq (I - \gamma P^{c\mu_k})^{-1} \big[ \gamma P^{\pi_k}(Q_k - Q^*) - \gamma P^{c\mu_k}(Q_k - Q^*) \big] \\
&= \gamma (I - \gamma P^{c\mu_k})^{-1} \big[ P^{\pi_k} - P^{c\mu_k} \big] (Q_k - Q^*), \\
&= A_k (Q_k - Q^*), \qquad \qquad (10)
\end{aligned}
$$

where $A_k := \gamma (I - \gamma P^{c\mu_k})^{-1} \big[ P^{\pi_k} - P^{c\mu_k} \big]$.

Now let us prove that $A_k$ has non-negative elements, whose sum over each row is at most $\gamma$. Let $e$ be the vector with 1-components. By rewriting $A_k$ as $\gamma \sum_{t \geq 0} \gamma^t (P^{c\mu_k})^t (P^{\pi_k} - P^{c\mu_k})$ and noticing that

$$
(P^{\pi_k} - P^{c\mu_k}) e(x,a) = \sum_{x'} \sum_{a'} p(x'|x,a) [\pi_k(a'|x') - c(a',x') \mu_k(a'|x')] \geq 0, \qquad (11)
$$

it is clear that all elements of $A_k$ are non-negative. We have

$$
\begin{aligned}
A_k e &= \gamma \sum_{t \geq 0} \gamma^t (P^{c\mu_k})^t \big[ P^{\pi_k} - P^{c\mu_k} \big] e \\
&= \gamma \sum_{t \geq 0} \gamma^t (P^{c\mu_k})^t e - \sum_{t \geq 0} \gamma^{t+1} (P^{c\mu_k})^{t+1} e \\
&= e - (1 - \gamma) \sum_{t \geq 0} \gamma^t (P^{c\mu_k})^t e \\
&\leq \gamma e, \qquad \qquad (12)
\end{aligned}
$$

(since $\sum_{t\geq 0}\gamma^t(P^{c\mu_k})^t e \geq e$). Thus $A_k$ has non-negative elements, whose sum over each row, is at most $\gamma$. We deduce from (10) that $Q_{k+1} - Q^*$ is upper-bounded by a sub-convex combination of components of $Q_k - Q^*$; the sum of their coefficients is at most $\gamma$. Thus

$$Q_{k+1} - Q^* \leq \gamma \|Q_k - Q^*\| e. \tag{13}$$

**Lower bound on $Q_{k+1} - Q^*$.** We have

$$
\begin{aligned}
Q_{k+1} &= Q_k + (I - \gamma P^{c\mu_k})^{-1}(\mathcal{T}^{\pi_k}Q_k - Q_k) \\
&= Q_k + \sum_{i\geq 0}\gamma^i(P^{c\mu_k})^i(\mathcal{T}^{\pi_k}Q_k - Q_k) \\
&= \mathcal{T}^{\pi_k}Q_k + \sum_{i\geq 1}\gamma^i(P^{c\mu_k})^i(\mathcal{T}^{\pi_k}Q_k - Q_k) \\
&= \mathcal{T}^{\pi_k}Q_k + \gamma P^{c\mu_k}(I - \gamma P^{c\mu_k})^{-1}(\mathcal{T}^{\pi_k}Q_k - Q_k). \tag{14}
\end{aligned}
$$

Now, from the definition of $\varepsilon_k$ we have $\mathcal{T}^{\pi_k}Q_k \geq \mathcal{T}Q_k - \varepsilon_k\|Q_k\| \geq \mathcal{T}^{\pi^*}Q_k - \varepsilon_k\|Q_k\|$, thus

$$
\begin{aligned}
Q_{k+1} - Q^* &= Q_{k+1} - \mathcal{T}^{\pi_k}Q_k + \mathcal{T}^{\pi_k}Q_k - \mathcal{T}^{\pi^*}Q_k + \mathcal{T}^{\pi^*}Q_k - \mathcal{T}^{\pi^*}Q^* \\
&\geq Q_{k+1} - \mathcal{T}^{\pi_k}Q_k + \gamma P^{\pi^*}(Q_k - Q^*) - \varepsilon_k\|Q_k\| e
\end{aligned}
$$

Using (14) we derive the lower bound:

$$Q_{k+1} - Q^* \geq \gamma P^{c\mu_k}(I - \gamma P^{c\mu_k})^{-1}(\mathcal{T}^{\pi_k}Q_k - Q_k) + \gamma P^{\pi^*}(Q_k - Q^*) - \varepsilon_k\|Q_k\|. \tag{15}$$

**Lower bound on $\mathcal{T}^{\pi_k}Q_k - Q_k$.** By hypothesis, $(\pi_k)$ is increasingly greedy w.r.t. $(Q_k)$, thus

$$
\begin{aligned}
\mathcal{T}^{\pi_{k+1}}Q_{k+1} - Q_{k+1} &\geq \mathcal{T}^{\pi_k}Q_{k+1} - Q_{k+1} \\
&= \mathcal{T}^{\pi_k}\mathcal{R}_k Q_k - \mathcal{R}_k Q_k \\
&= r + (\gamma P^{\pi_k} - I)\mathcal{R}_k Q_k \\
&= r + (\gamma P^{\pi_k} - I)\big[Q_k + (I - \gamma P^{c\mu_k})^{-1}(\mathcal{T}^{\pi_k}Q_k - Q_k)\big] \\
&= \mathcal{T}^{\pi_k}Q_k - Q_k + (\gamma P^{\pi_k} - I)(I - \gamma P^{c\mu_k})^{-1}(\mathcal{T}^{\pi_k}Q_k - Q_k) \\
&= \gamma\big[P^{\pi_k} - P^{c\mu_k}\big](I - \gamma P^{c\mu_k})^{-1}(\mathcal{T}^{\pi_k}Q_k - Q_k) \\
&= B_k(\mathcal{T}^{\pi_k}Q_k - Q_k), \tag{16}
\end{aligned}
$$

where $B_k := \gamma[P^{\pi_k} - P^{c\mu_k}](I - \gamma P^{c\mu_k})^{-1}$. Since $P^{\pi_k} - P^{c\mu_k}$ has non-negative elements (as proven in (11)) as well as $(I - \gamma P^{c\mu_k})^{-1}$, then $B_k$ has non-negative elements as well. Thus

$$\mathcal{T}^{\pi_k}Q_k - Q_k \geq B_{k-1}B_{k-2}\ldots B_0(\mathcal{T}^{\pi_0}Q_0 - Q_0) \geq 0,$$

since we assumed $T^{\pi_0}Q_0 - Q_0 \geq 0$. Thus (15) implies that

$$Q_{k+1} - Q^* \geq \gamma P^{\pi^*}(Q_k - Q^*) - \varepsilon_k\|Q_k\|.$$

and combining the above with (13) we deduce

$$\|Q_{k+1} - Q^*\| \leq \gamma\|Q_k - Q^*\| + \varepsilon_k\|Q_k\|.$$

Now assume that $\varepsilon_k \to 0$. We first deduce that $Q_k$ is bounded. Indeed as soon as $\varepsilon_k < (1-\gamma)/2$, we have

$$\|Q_{k+1}\| \leq \|Q^*\| + \gamma\|Q_k - Q^*\| + \frac{1-\gamma}{2}\|Q_k\| \leq (1+\gamma)\|Q^*\| + \frac{1+\gamma}{2}\|Q_k\|.$$

Thus $\limsup\|Q_k\| \leq \frac{1+\gamma}{1-(1+\gamma)/2}\|Q^*\|$. Since $Q_k$ is bounded, we deduce that $\limsup Q_k = Q^*$. $\qquad\square$

# D   Proof of Theorem 3

We first prove convergence of the general online algorithm.

**Theorem 4.** *Consider the algorithm*

$$Q_{k+1}(x,a) = (1 - \alpha_k(x,a))Q_k(x,a) + \alpha_k(x,a)(\mathcal{R}_k Q_k(x,a) + \omega_k(x,a) + \upsilon_k(x,a)), \quad (17)$$

*and assume that (1) $\omega_k$ is a centered, $\mathcal{F}_k$-measurable noise term of bounded variance, and (2) $\upsilon_k$ is bounded from above by $\theta_k(\|Q_k\| + 1)$, where $(\theta_k)$ is a random sequence that converges to 0 a.s. Then, under the same assumptions as in Theorem 3, we have that $Q_k \to Q^*$ almost surely.*

*Proof.* We write $\mathcal{R}$ for $\mathcal{R}_k$. Let us prove the result in three steps.

**Upper bound on $\mathcal{R}Q_k - Q^*$.** The first part of the proof is similar to the proof of (13), so we have

$$\mathcal{R}Q_k - Q^* \leq \gamma\|Q_k - Q^*\|e. \quad (18)$$

**Lower bound on $\mathcal{R}Q_k - Q^*$.** Again, similarly to (15) we have

$$
\begin{aligned}
\mathcal{R}Q_k - Q^* \geq{}& \gamma\lambda P^{\pi_k \wedge \mu_k}(I - \gamma\lambda P^{\pi_k \wedge \mu_k})^{-1}(\mathcal{T}^{\pi_k}Q_k - Q_k) \\
&+\gamma P^{\pi^*}(Q_k - Q^*) - \varepsilon_k\|Q_k\|.
\end{aligned}
\quad (19)
$$

**Lower-bound on $\mathcal{T}^{\pi_k}Q_k - Q_k$.** Since the sequence of policies $(\pi_k)$ is increasingly greedy w.r.t. $(Q_k)$, we have

$$
\begin{aligned}
\mathcal{T}^{\pi_{k+1}}Q_{k+1} - Q_{k+1} \geq{}& \mathcal{T}^{\pi_k}Q_{k+1} - Q_{k+1} \\
={}& (1 - \alpha_k)\mathcal{T}^{\pi_k}Q_k + \alpha_k\mathcal{T}^{\pi_k}(\mathcal{R}Q_k + \omega_k + \upsilon_k) - Q_{k+1} \\
={}& (1 - \alpha_k)(\mathcal{T}^{\pi_k}Q_k - Q_k) + \alpha_k\left[\mathcal{T}^{\pi_k}\mathcal{R}Q_k - \mathcal{R}Q_k + \omega_k' + \upsilon_k'\right],
\end{aligned}
\quad (20)
$$

where $\omega_k' := (\gamma P^{\pi_k} - I)\omega_k$ and $\upsilon_k' := (\gamma P^{\pi_k} - I)\upsilon_k$. It is easy to see that both $\omega_k'$ and $\upsilon_k'$ continue to satisfy the assumptions on $\omega_k$, and $\upsilon_k$. Now, from the definition of the $\mathcal{R}$ operator, we have

$$
\begin{aligned}
\mathcal{T}^{\pi_k}\mathcal{R}Q_k - \mathcal{R}Q_k ={}& r + (\gamma P^{\pi_k} - I)\mathcal{R}Q_k \\
={}& r + (\gamma P^{\pi_k} - I)\left[Q_k + (I - \gamma\lambda P^{\pi_k \wedge \mu_k})^{-1}(\mathcal{T}^{\pi_k}Q_k - Q_k)\right] \\
={}& \mathcal{T}^{\pi_k}Q_k - Q_k + (\gamma P^{\pi_k} - I)(I - \gamma\lambda P^{\pi_k \wedge \mu_k})^{-1}(\mathcal{T}^{\pi_k}Q_k - Q_k) \\
={}& \gamma(P^{\pi_k} - \lambda P^{\pi_k \wedge \mu_k})(I - \gamma\lambda P^{\pi_k \wedge \mu_k})^{-1}(\mathcal{T}^{\pi_k}Q_k - Q_k).
\end{aligned}
$$

Using this equality into (20) and writing $\xi_k := \mathcal{T}^{\pi_k}Q_k - Q_k$, we have

$$\xi_{k+1} \geq (1 - \alpha_k)\xi_k + \alpha_k\left[B_k\xi_k + \omega_k' + \upsilon_k'\right], \quad (21)$$

where $B_k := \gamma(P^{\pi_k} - \lambda P^{\pi_k \wedge \mu_k})(I - \gamma\lambda P^{\pi_k \wedge \mu_k})^{-1}$. The matrix $B_k$ is non-negative but may not be a contraction mapping (the sum of its components per row may be larger than 1). Thus we cannot directly apply Proposition 4.5 of Bertsekas and Tsitsiklis (1996). However, as we have seen in the proof of Theorem 2, the matrix $A_k := \gamma(I - \gamma\lambda P^{\pi_k \wedge \mu_k})^{-1}(P^{\pi_k} - \lambda P^{\pi_k \wedge \mu_k})$ is a $\gamma$-contraction mapping. So now we relate $B_k$ to $A_k$ using our assumption that $P^{\pi_k}$ and $P^{\pi_k \wedge \mu_k}$ commute asymptotically, i.e. $\|P^{\pi_k}P^{\pi_k \wedge \mu_k} - P^{\pi_k \wedge \mu_k}P^{\pi_k}\| = \eta_k$ with $\eta_k \to 0$. For any (sub)-transition matrices $U$ and $V$, we have

$$
\begin{aligned}
U(I - \lambda\gamma V)^{-1} ={}& \sum_{t \geq 0}(\lambda\gamma)^t U V^t \\
={}& \sum_{t \geq 0}(\lambda\gamma)^t\left[\sum_{s=0}^{t-1}V^s(UV - VU)V^{t-s-1} + V^t U\right] \\
={}& (I - \lambda\gamma V)^{-1}U + \sum_{t \geq 0}(\lambda\gamma)^t\sum_{s=0}^{t-1}V^s(UV - VU)V^{t-s-1}.
\end{aligned}
$$

Replacing $U$ by $P^{\pi_k}$ and $V$ by $P^{\pi_k \wedge \mu_k}$, we deduce

$$\|B_k - A_k\| \leq \gamma\sum_{t \geq 0}t(\lambda\gamma)^t\eta_k = \gamma\frac{1}{(1 - \lambda\gamma)^2}\eta_k.$$

Thus, from (21),

$$\xi_{k+1} \geq (1 - \alpha_k)\xi_k + \alpha_k \big[ A_k \xi_k + \omega'_k + \upsilon''_k \big], \tag{22}$$

where $\upsilon''_k := \upsilon'_k + \gamma \sum_{t \geq 0} t(\lambda\gamma)^t \eta_k \|\xi_k\|$ continues to satisfy the assumptions on $\upsilon_k$ (since $\eta_k \to 0$).

Now, let us define another sequence $\xi'_k$ as follows: $\xi'_0 = \xi_0$ and

$$\xi'_{k+1} = (1 - \alpha_k)\xi'_k + \alpha_k(A_k \xi'_k + \omega'_k + \upsilon''_k).$$

We can now apply Proposition 4.5 of Bertsekas and Tsitsiklis (1996) to the sequence $(\xi'_k)$. The matrices $A_k$ are non-negative, and the sum of their coefficients per row is bounded by $\gamma$, see (12), thus $A_k$ are $\gamma$-contraction mappings and have the same fixed point which is 0. The noise $\omega'_k$ is centered and $\mathcal{F}_k$-measurable and satisfies the bounded variance assumption, and $\upsilon''_k$ is bounded above by $(1 + \gamma)\theta'_k(\|Q_k\| + 1)$ for some $\theta'_k \to 0$. Thus $\lim_k \xi'_k = 0$ almost surely.

Now, it is straightforward to see that $\xi_k \geq \xi'_k$ for all $k \geq 0$. Indeed by induction, let us assume that $\xi_k \geq \xi'_k$. Then

$$
\begin{aligned}
\xi_{k+1} &\geq (1 - \alpha_k)\xi_k + \alpha_k(A_k \xi_k + \omega'_k + \upsilon''_k) \\
&\geq (1 - \alpha_k)\xi'_k + \alpha_k(A_k \xi'_k + \omega'_k + \upsilon''_k) \\
&= \xi'_{k+1},
\end{aligned}
$$

since all elements of the matrix $A_k$ are non-negative. Thus we deduce that

$$\liminf_{k \to \infty} \xi_k \geq \lim_{k \to \infty} \xi'_k = 0 \tag{23}$$

**Conclusion.** Using (23) in (19) we deduce the lower bound:

$$\liminf_{k \to \infty} \mathcal{R}Q_k - Q^* \geq \liminf_{k \to \infty} \gamma P^{\pi^*}(Q_k - Q^*), \tag{24}$$

almost surely. Now combining with the upper bound (18) we deduce that

$$\|\mathcal{R}Q_k - Q^*\| \leq \gamma\|Q_k - Q^*\| + O(\varepsilon_k\|Q_k\|) + O(\xi_k).$$

The last two terms can be incorporated to the $\upsilon_k(x, a)$ and $\omega_k(x, a)$ terms, respectively; we thus again apply Proposition 4.5 of Bertsekas and Tsitsiklis (1996) to the sequence $(Q_k)$ defined by (17) and deduce that $Q_k \to Q^*$ almost surely. $\qquad\square$

It remains to rewrite the update (7) in the form of (17), in order to apply Theorem 4.

Let $z^k_{s,t}$ denote the accumulating trace (Sutton and Barto, 1998):

$$z^k_{s,t} := \sum_{j=s}^{t} \gamma^{t-j} \Big( \prod_{i=j+1}^{t} c_i \Big) \mathbb{I}\{(x_j, a_j) = (x_s, a_s)\}.$$

Let us write $Q^o_{k+1}(x_s, a_s)$ to emphasize the online setting. Then (7) can be written as

$$Q^o_{k+1}(x_s, a_s) \leftarrow Q^o_k(x_s, a_s) + \alpha_k(x_s, a_s) \sum_{t \geq s} \delta^{\pi_k}_t z^k_{s,t}, \tag{25}$$

$$\delta^{\pi_k}_t := r_t + \gamma \mathbb{E}_{\pi_k} Q^o_k(x_{t+1}, \cdot) - Q^o_k(x_t, a_t),$$

Using our assumptions on finite trajectories, and $c_i \leq 1$, we can show that:

$$\mathbb{E}\Big[ \sum_{t \geq s} z^k_{s,t} \big| \mathcal{F}_k \Big] < \mathbb{E}\big[ T^2_k | \mathcal{F}_k \big] < \infty \tag{26}$$

where $T_k$ denotes trajectory length. Now, let $D_k := D_k(x_s, a_s) := \sum_{t \geq s} \mathbb{P}\{(x_t, a_t) = (x_s, a_s)\}$. Then, using (26), we can show that the total update is bounded, and rewrite

$$\mathbb{E}_{\mu_k}\Big[ \sum_{t \geq s} \delta^{\pi_k}_t z^k_{s,t} \Big] = D_k(x_s, a_s)\big( \mathcal{R}_k Q_k(x_s, a_s) - Q(x_s, a_s) \big).$$

Finally, using the above, and writing $\alpha_k = \alpha_k(x_s, a_s)$, (25) can be rewritten in the desired form:

$$Q_{k+1}^o(x_s, a_s) \leftarrow (1 - \tilde{\alpha}_k)Q_k^o(x_s, a_s) + \tilde{\alpha}_k\big(\mathcal{R}_k Q_k^o(x_s, a_s) + \omega_k(x_s, a_s) + \upsilon_k(x_s, a_s)\big), \quad (27)$$

$$\omega_k(x_s, a_s) := (D_k)^{-1}\left(\sum_{t \geq s} \delta_t^{\pi_k} z_{s,t}^k - \mathbb{E}_{\mu_k}\left[\sum_{t \geq s} \delta_t^{\pi_k} z_{s,t}^k\right]\right),$$

$$\upsilon_k(x_s, a_s) := (\tilde{\alpha}_k)^{-1}\big(Q_{k+1}^o(x_s, a_s) - Q_{k+1}(x_s, a_s)\big),$$

$$\tilde{\alpha}_k := \alpha_k D_k.$$

It can be shown that the variance of the noise term $\omega_k$ is bounded, using (26) and the fact that the reward function is bounded. It follows from Assumptions 1-3 that the modified stepsize sequence $(\tilde{\alpha}_k)$ satisfies the conditions of Assumption 1. The second noise term $\upsilon_k(x_s, a_s)$ measures the difference between online iterates and the corresponding offline values, and can be shown to satisfy the required assumption analogously to the argument in the proof of Prop. 5.2 in Bertsekas and Tsitsiklis (1996). The proof relies on the eligibility coefficients (26) and rewards being bounded, the trajectories being finite, and the conditions on the stepsizes being satisfied.

We can thus apply Theorem 4 to (27), and conclude that the iterates $Q_k^o \to Q^*$ as $k \to \infty$, w.p. 1.

# E  Asymptotic commutativity of $P^{\pi_k}$ and $P^{\pi_k \wedge \mu_k}$

**Lemma 4.** *Let $(\pi_k)$ and $(\mu_k)$ two sequences of policies. If there exists $\alpha$ such that for all $x, a$,*

$$\min(\pi_k(a|x), \mu_k(a|x)) = \alpha\pi_k(a|x) + o(1), \quad (28)$$

*then the transition matrices $P^{\pi_k}$ and $P^{\pi_k \wedge \mu_k}$ asymptotically commute:* $\|P^{\pi_k}P^{\pi_k \wedge \mu_k} - P^{\pi_k \wedge \mu_k}P^{\pi_k}\| = o(1)$.

*Proof.* For any $Q$, we have

$$(P^{\pi_k}P^{\pi_k \wedge \mu_k})Q(x, a) = \sum_y p(y|x, a)\sum_b \pi_k(b|y)\sum_z p(z|y, b)\sum_c (\pi_k \wedge \mu_k)(c|z)Q(z, c)$$

$$= \alpha\sum_y p(y|x, a)\sum_b \pi_k(b|y)\sum_z p(z|y, b)\sum_c \pi_k(c|z)Q(z, c) + \|Q\|o(1)$$

$$= \sum_y p(y|x, a)\sum_b (\pi_k \wedge \mu_k)(b|y)\sum_z p(z|y, b)\sum_c \pi_k(c|z)Q(z, c) + \|Q\|o(1)$$

$$= (P^{\pi_k \wedge \mu_k}P^{\pi_k})Q(x, a) + \|Q\|o(1). \qquad \square$$

**Lemma 5.** *Let $(\pi_{Q_k})$ a sequence of (deterministic) greedy policies w.r.t. a sequence $(Q_k)$. Let $(\pi_k)$ a sequence of policies that are $\varepsilon_k$ away from $(\pi_{Q_k})$, in the sense that, for all $x$,*

$$\|\pi_k(\cdot|x) - \pi_{Q_k}(x)\|_1 := 1 - \pi_k(\pi_{Q_k}(x)|x) + \sum_{a \neq \pi_{Q_k}(x)} \pi_k(a|x) \leq \varepsilon_k.$$

*Let $(\mu_k)$ a sequence of policies defined by:*

$$\mu_k(a|x) = \frac{\alpha\mu(a|x)}{1 - \mu(\pi_{Q_k}(x)|x)}\mathbb{I}\{a \neq \pi_{Q_k}(x)\} + (1 - \alpha)\mathbb{I}\{a = \pi_{Q_k}(x)\}, \quad (29)$$

*for some arbitrary policy $\mu$ and $\alpha \in [0, 1]$. Assume $\varepsilon_k \to 0$. Then the transition matrices $P^{\pi_k}$ and $P^{\pi_k \wedge \mu_k}$ asymptotically commute.*

*Proof.* The intuition is that asymptotically $\pi_k$ gets very close to the deterministic policy $\pi_{Q_k}$. In that case, the minimum distribution $(\pi_k \wedge \mu_k)(\cdot|x)$ puts a mass close to $1 - \alpha$ on the greedy action $\pi_{Q_k}(x)$, and no mass on other actions, thus $(\pi_k \wedge \mu_k)$ gets very close to $(1 - \alpha)\pi_k$, and Lemma 4 applies (with multiplicative constant $1 - \alpha$).

Indeed, from our assumption that $\pi_k$ is $\varepsilon$-away from $\pi_{Q_k}$ we have:

$$\pi_k(\pi_{Q_k}(x)|x) \geq 1 - \varepsilon_k, \text{ and } \pi_k(a \neq \pi_{Q_k}(x)|x) \leq \varepsilon_k.$$

We deduce that

$$
\begin{aligned}
(\pi_k \wedge \mu_k)(\pi_{Q_k}(x)|x) &= \min(\pi_k(\pi_{Q_k}(x)|x), 1 - \alpha) \\
&= 1 - \alpha + O(\varepsilon_k) \\
&= (1 - \alpha)\pi_k(\pi_{Q_k}(x)|x) + O(\varepsilon_k),
\end{aligned}
$$

and

$$
\begin{aligned}
(\pi_k \wedge \mu_k)(a \neq \pi_{Q_k}(x)|x) &= O(\varepsilon_k) \\
&= (1 - \alpha)\pi_k(a|x) + O(\varepsilon_k).
\end{aligned}
$$

Thus Lemma 4 applies (with a multiplicative constant $1 - \alpha$) and $P^{\pi_k}$ and $P^{\pi_k \wedge \mu_k}$ asymptotically commute. $\qquad\square$

## F  Experimental Methods

Although our experiments' learning problem closely matches the DQN setting used by Mnih et al. (2015) (i.e. single-thread off-policy learning with large replay memory), we conducted our trials in the multi-threaded, CPU-based framework of Mnih et al. (2016), obtaining ample result data from affordable CPU resources. Key differences from the DQN are as follows. Sixteen threads with private environment instances train simultaneously; each infers with and finds gradients w.r.t. a local copy of the network parameters; gradients then update a "master" parameter set and local copies are refreshed. Target network parameters are simply shared globally. Each thread has private replay memory holding 62,500 transitions (1/16[th] of DQN's total replay capacity). The optimizer is unchanged from (Mnih et al., 2016): "Shared RMSprop" with step size annealing to 0 over $3 \times 10^8$ environment frames (summed over threads). Exploration parameter ($\varepsilon$) behaviour differs slightly: every 50,000 frames, threads switch randomly (probability 0.3, 0.4, and 0.3 respectively) between three schedules (anneal $\varepsilon$ from 1 to 0.5, 0.1, or 0.01 over 250,000 frames), starting new schedules at the intermediate positions where they left old ones.[1]

Our experiments comprise 60 Atari 2600 games in ALE (Bellemare et al., 2013), with "life" loss treated as episode termination. The control, minibatched (64 transitions/minibatch) one-step Q-learning as in (Mnih et al., 2015), shows performance comparable to DQN in our multi-threaded setup. Retrace, TB, and $Q^*$ runs use minibatches of four 16-step sequences (again 64 transitions/minibatch) and the current exploration policy as the target policy $\pi$. All trials clamp rewards into $[-1, 1]$. In the control, Q-function targets are clamped into $[-1, 1]$ prior to gradient calculation; analogous quantities in the multi-step algorithms are clamped into $[-1, 1]$, then scaled (divided by) the sequence length. Coarse, then fine logarithmic parameter sweeps on the games *Asterix*, *Breakout*, *Enduro*, *Freeway*, *H.E.R.O*, *Pong*, *Q\*bert*, and *Seaquest* yielded step sizes of 0.0000439 and 0.0000912, and RMSprop regularization parameters of 0.001 and 0.0000368, for control and multi-step algorithms respectively. Reported performance averages over four trials with different random seeds for each experimental configuration.

### F.1  Algorithmic Performance in Function of $\lambda$

We compared our algorithms for different values of $\lambda$, using the DQN score as a baseline. As before, for each $\lambda$ we compute the inter-algorithm scores on a per-game basis. We then averaged the inter-algorithm scores across games to produce Table 2 (see also Figure 2 for a visual depiction). We first remark that Retrace always achieve a score higher than TB, demonstrating that it is efficient in the sense of Section 2. Next, we note that $Q^*$ performs best for small values of $\lambda$, but begins to fail for values above $\lambda = 0.5$. In this sense, it is also not safe. This is particularly problematic as the safe threshold of $\lambda$ is likely to be problem-dependent. Finally, there is no setting of $\lambda$ for which Retrace performs particularly poorly; for high values of $\lambda$, it achieves close to the top score in most games. For Retrace($\lambda$) it makes sense to use a values $\lambda = 1$ (at least in deterministic environments) as the trace cutting effect required in off-policy learning is taken care of by the use of the $\min(1, \pi/\mu)$ coefficient. On the contrary, $Q^*(\lambda)$ only relies on a value $\lambda < 1$ to take care of cutting traces for off-policy data.

| $\lambda$ | DQN | TB | Retrace | $Q^*$ |
|---|---|---|---|---|
| 0.0 | 0.5071 | 0.5512 | 0.4288 | 0.4487 |
| 0.1 | 0.4752 | 0.2798 | 0.5046 | 0.651 |
| 0.3 | 0.3634 | 0.268 | 0.5159 | 0.7734 |
| 0.5 | 0.2409 | 0.4105 | 0.5098 | 0.8419 |
| 0.7 | 0.3712 | 0.4453 | 0.6762 | 0.5551 |
| 0.9 | 0.7256 | 0.7753 | 0.9034 | 0.02926 |
| 1.0 | 0.6839 | 0.8158 | 0.8698 | 0.04317 |

Table 2: Average inter-algorithm scores for each value of $\lambda$. The DQN scores are fixed across different $\lambda$, but the corresponding inter-algorithm scores varies depending on the worst and best performer within each $\lambda$.

Figure 2: Average inter-algorithm scores for each value of $\lambda$. The DQN scores are fixed across different $\lambda$, but the corresponding inter-algorithm scores varies depending on the worst and best performer within each $\lambda$. **Note that average scores are not directly comparable across different values of $\lambda$.**

|  | Tree-backup($\lambda$) | Retrace($\lambda$) | DQN | Q*($\lambda$) |
|---|---|---|---|---|
| ALIEN | 2508.62 | **3109.21** | 2088.81 | 154.35 |
| AMIDAR | 1221.00 | **1247.84** | 772.30 | 16.04 |
| ASSAULT | 7248.08 | **8214.76** | 1647.25 | 260.95 |
| ASTERIX | **29294.76** | 28116.39 | 10675.57 | 285.44 |
| ASTEROIDS | 1499.82 | **1538.25** | 1403.19 | 308.70 |
| ATLANTIS | **2115949.75** | 2110401.90 | 1712671.88 | 3667.18 |
| BANK HEIST | **808.31** | 797.36 | 549.35 | 1.70 |
| BATTLE ZONE | 22197.96 | **23544.08** | 21700.01 | 3278.93 |
| BEAM RIDER | 15931.60 | **17281.24** | 8053.26 | 621.40 |
| BERZERK | 967.29 | **972.67** | 627.53 | 247.80 |
| BOWLING | 40.96 | **47.92** | 37.82 | 15.16 |
| BOXING | 91.00 | 93.54 | **95.17** | -29.25 |
| BREAKOUT | 288.71 | 298.75 | **332.67** | 1.21 |
| CARNIVAL | **4691.73** | 4633.77 | 4637.86 | 353.10 |
| CENTIPEDE | 1199.46 | 1715.95 | 1037.95 | **3783.60** |
| CHOPPER COMMAND | 6193.28 | **6358.81** | 5007.32 | 534.83 |
| CRAZY CLIMBER | **115345.95** | 114991.29 | 111918.64 | 1136.21 |
| DEFENDER | 32411.77 | **33146.83** | 13349.26 | 1838.76 |
| DEMON ATTACK | 68148.22 | **79954.88** | 8585.17 | 310.45 |
| DOUBLE DUNK | **-1.32** | -6.78 | -5.74 | -23.63 |
| ELEVATOR ACTION | 1544.91 | 2396.05 | **14607.10** | 930.38 |
| ENDURO | 1115.00 | **1216.47** | 938.36 | 12.54 |
| FISHING DERBY | 22.22 | **27.69** | 15.14 | -98.58 |
| FREEWAY | **32.13** | 32.13 | 31.07 | 9.86 |
| FROSTBITE | 960.30 | 935.42 | **1124.60** | 45.07 |
| GOPHER | 13666.33 | **14110.94** | 11542.46 | 50.59 |
| GRAVITAR | 30.18 | 29.04 | **271.40** | 13.14 |
| H.E.R.O. | **25048.33** | 21989.46 | 17626.90 | 12.48 |
| ICE HOCKEY | **-3.84** | -5.08 | -4.36 | -15.68 |
| JAMES BOND | 560.88 | 641.51 | **705.55** | 21.71 |
| KANGAROO | 11755.01 | **11896.25** | 4101.92 | 178.23 |
| KRULL | **9509.83** | 9485.39 | 7728.66 | 429.26 |
| KUNG-FU MASTER | 25338.05 | **26695.19** | 17751.73 | 39.99 |
| MONTEZUMA'S REVENGE | **0.79** | 0.18 | 0.10 | 0.00 |
| MS. PAC-MAN | 2461.10 | **3208.03** | 2654.97 | 298.58 |
| NAME THIS GAME | **11358.81** | 11160.15 | 10098.85 | 1311.73 |
| PHOENIX | 13834.27 | **15637.88** | 9249.38 | 107.41 |
| PITFALL | **-37.74** | -43.85 | -392.63 | -121.99 |
| POOYAN | 5283.69 | **5661.92** | 3301.69 | 98.65 |
| PONG | **20.25** | 20.20 | 19.31 | -20.99 |
| PRIVATE EYE | 73.44 | **87.36** | 44.73 | -147.49 |
| Q*BERT | 13617.24 | **13700.25** | 12412.85 | 114.84 |
| RIVER RAID | 14457.29 | **15365.61** | 10329.58 | 922.13 |
| ROAD RUNNER | 34396.52 | 32843.09 | **50523.75** | 418.62 |
| ROBOTANK | 36.07 | 41.18 | **49.20** | 5.77 |
| SEAQUEST | 3557.09 | 2914.00 | **3869.30** | 175.29 |
| SKIING | -25055.94 | -25235.75 | -25254.43 | **-24179.71** |
| SOLARIS | 1178.05 | 1135.51 | **1258.02** | 674.58 |
| SPACE INVADERS | **6096.21** | 5623.34 | 2115.80 | 227.39 |
| STAR GUNNER | 66369.18 | **74016.10** | 42179.52 | 266.15 |
| SURROUND | **-5.48** | -6.04 | -8.17 | -9.98 |
| TENNIS | -1.73 | -0.30 | **13.67** | -7.37 |
| TIME PILOT | 8266.79 | **8719.19** | 8228.89 | 657.59 |
| TUTANKHAM | 164.54 | **199.25** | 167.22 | 2.68 |
| UP AND DOWN | 14976.51 | **18747.40** | 9404.95 | 530.59 |
| VENTURE | 10.75 | 22.84 | **30.93** | 0.09 |
| VIDEO PINBALL | 103486.09 | **228283.79** | 76691.75 | 6837.86 |
| WIZARD OF WOR | 7402.99 | **8048.72** | 612.52 | 189.43 |
| YAR'S REVENGE | 14581.65 | **26860.57** | 15484.03 | 1913.19 |
| ZAXXON | 12529.22 | **15383.11** | 8422.49 | 0.40 |
| Times Best | 16 | 30 | 12 | 2 |

Table 3: Final scores achieved by the different $\lambda$-return variants ($\lambda = 1$). Highlights indicate high scores.