[Reviews · NeurIPS 2016]

Reviewer 1

Summary

A general form of eligibility traces for Q-function evaluation and improvement (without function approximation) is proposed and analyzed. This framework encapsulates several known off-policy algorithms. In addition, a new algorithm is proposed that leverages the analysis for obtaining efficiency and low variance while still maintaining convergence. This algorithm is evaluated empirically in a deep RL setting with function approximation, where the experience replay database is used as the off-policy data.

Qualitative Assessment

This is a well-written and high quality paper, and I enjoyed reading it. This study unifies several previous algorithms under a general framework, and develops several novel theoretical contributions that shed light on rather old ideas. The analysis is elegant and non-trivial, and I believe that the techniques will be useful for future studies. The empirical evaluation is extensive (and somewhat formidable), and it is satisfying to see that a seemingly simple algorithmic modification, which stems from the theoretical understanding, also works in a large domain when most of the theoretical assumptions (such as no function approximation) don't hold. Minor comments / typos: 62: on the last equation, should be T^\pi Q - Q instead of T^\pi - Q 220: some typo in this sentence

Confidence in this Review

2-Confident (read it all; understood it all reasonably well)


Reviewer 2

Summary

This will be a seminal paper in reinforcement learning literature

Qualitative Assessment

This is a very good paper, however, due to space limitations, it is very dense to read, which can be understood.

Confidence in this Review

3-Expert (read the paper in detail, know the area, quite certain of my opinion)


Reviewer 3

Summary

The paper proves convergence for a class of off policy algorithms, settling an open problem in the affirmative.

Qualitative Assessment

The paper proves interesting results for a class off off policy algorithms, including Watkins Q lambda. I am not sure how correct the proofs are, particularly in the appendix. Theorem 3 sidesteps a contractibility issue but im not sure if the other assumptions of bertsekas apply. This issue should be made more prominent in the main text. I'm also not sure that the experiments show anything useful. First of all, they don't seem to be unbiased as lambda is effectively chosen on test data. Secondly the differences seem small. Minor note: it seems that theorem 1 only shows that Q is a fixed point for R but not uniqueness.

Confidence in this Review

3-Expert (read the paper in detail, know the area, quite certain of my opinion)


Reviewer 4

Summary

The paper introduces a new algorithm called Retrace(\lambda) that is meant to achieve a trade-off between efficiency and stability. In the process, it proves convergence of a large class of off-policy value function algorithms (including Q-learning with eligibility traces) in finite state spaces. The theoretical analysis is underpinned by a new general way of writing off-policy value function updates with eligibility traces (eq. 4). Experimental results show the Retrace algorithm performing better than DQN and the tree backup algorithm on the Atari domain.

Qualitative Assessment

***** UPDATE ***** I have lowered my scores for novelty and impact because I just came across the paper "High Confidence Off-Policy Evaluation" by Thomas, Theocharous and Ghavamzadeh (not referenced paper 602), which deals with some very similar issues. In particular, it bounds the performance of off-policy importance sampling as a function of a truncation coefficient, and discusses how to choose that coefficient based on the bound they propose. The lack of a discussion of the relationship to that work makes paper 602 considerably weaker in my opinion. I would still lean towards acceptance, but only as a poster. ******************* The paper is a great starting point for a theoretical understanding of current off-policy value-based methods and for developing new ones. Analyzing the convergence of the general-form off-policy updates in Equation 4 is novel and important. The theory is limited to finite state spaces (something that should be stated in the abstract) for discounted MDPs, but the empirical results show that the new Retrace algorithm can perform well in conjunction with value function approximation. The theoretical results are sound as far as I can tell. Retrace as the combination of truncated importance sampling and eligibility traces is a new method, however the authors should cite existing work on reducing the variance of importance sampling by truncating coefficients, e.g. "Truncated Importance Sampling" (Ionides, 2008). The paper suffers in clarity to some extent because a lot of content is being crammed within the bounds of a conference paper. In particular, some of the theory takes a while to read through as several steps are condensed into one, and the experimental results at the end feel a bit rushed. Also at several points the text is begging for more discussion (e.g. about the asymptotic commutativity assumption vs GLIE, or why limit the upper bound for c_s at 1 as opposed to some other value). Perhaps a good solution would be for the paper to only include the theory and discussion, and to reference an extended version containing a more in-depth examination of the experimental results - my feeling is that a few years down the road the theory in this paper will be of a lot more interest than the experiments. Several comments regarding clarity (some of them minor): - in the equation above line 62, the operator definition is missing something (it becomes equal to zero when lambda=1) - line 131: i would say that it is more obvious that Q^\pi is the fixed point from Lemma 1 than from Equation 4 - typo in lines 219-220 - the equation after line 262 doesn't look like a Q-function target, but rather as part of a gradient update - line 270: Figure 1 does not show that Q* diverges for lambda=1

Confidence in this Review

2-Confident (read it all; understood it all reasonably well)


Reviewer 5

Summary

Authors propose a new off-policy reinforcement learning algorithm called Trace. It combines the advantages of previously proposed algorithms, including Importance Sampling, off-policy Q and Tree-Backup.

Qualitative Assessment

Article is well-reasoned and the contribution seems relevant.

Confidence in this Review

1-Less confident (might not have understood significant parts)


Reviewer 6

Summary

The purpose of this paper is to introduce a return-based RL algorithm that has useful properties that other trace-based RL algorithms do not have. The paper includes proofs that show that that this approach converges. Additionally, empirical results show that this approach works well.

Qualitative Assessment

================== Technical quality ================== The paper includes proofs that show the approach is sound. Additionally, empirical results show the approach can work well. However, more analysis could be done for the results. For example, although retrace does better than TB, the difference between the scores for individual games was not that large. This brings into question how important the efficiency condition is. Additionally, one reason deep RL uses random samples from the replay is to decorrelate the data. Does sampling entire episodes have any negative effect on the performance? ================== Novelty ================== This approach combined properties of other trace-based methods to develop a method that had low variance, was "safe", and was efficient. The combination of those properties is novel, however each of the other approaches had two of these properties, and so the combination did not seem too groundbreaking or significant. Still, it is clearly a notable improvement. ================== Potential Impact ================== This work appears to perform better than other trace-based approaches and has a large advantage over Q-learning. It seems simple enough to use retrace with DQN or other methods. Many researchers could use this method to improve the performance of their algorithms. ================== Clarity ================== Most of the points in the paper were clearly explained. Line 52 was potentially confusing. It seems incorrect to call the reward a Q-function. Even if they both are a function of (s,a) and return a scalar value, they represent different things. Also, the paper could be improved by including a conclusion. ================== Other points ================== Line 153: greeedy -> greedy Line 217: improtant -> important Lines 219-220: Remove "prove online conver" Line 235: Missing period after "Increasingly greedy policies" Line 236: that -> than

Confidence in this Review

2-Confident (read it all; understood it all reasonably well)